# Development and external validation of prediction algorithms to improve early diagnosis of cancer

J. Hippisley-Cox [1] ✉ & CA Coupland[2]

Cancer prediction algorithms are used in the UK to identify individuals at high probability of having a current, as yet undiagnosed cancer with the intention of improving early diagnosis and treatment. Here we develop and externally validate two diagnostic prediction algorithms to estimate the probability of having cancer for 15 cancer types. The first incorporates multiple predictors including age, sex, deprivation, smoking, alcohol, family history, medical diagnoses and symptoms (both general and cancer-specific symptoms). The second additionally includes commonly used blood tests (full blood count and liver function tests). We use multinomial logistic regression to develop separate equations in men and women to predict the absolute probability of 15 cancer types using a population of 7.46 million adults aged 18 to 84 years in England. We evaluate performance in two separate validation cohorts (total 2.64 million patients in England and 2.74 million from Scotland, Wales and Northern Ireland). The models have improved performance compared with existing models with improved discrimination, calibration, sensitivity and net benefit. These algorithms provide superior prediction estimates in the UK compared with existing scores and could lead to better clinical decision-making and potentially earlier diagnosis of cancer.

The UK has some of the poorest clinical outcomes for cancer survival in the developed world[1]. This is probably explained by differences in cancer stage at diagnosis and timely access to effective treatments[1]. Delays in early diagnosis may be due to a combination of delays in presentation, investigation or referral[2,3]. In 2011, the UK National Health Service (NHS) published a cancer strategy with a target to diagnose 75% of cancers at a curable stage (i.e., stage 1 or 2)[2]. The strategy highlighted the potential to use predictive algorithms to improve early diagnosis in primary care[2]. This led to the development and validation of cancer prediction algorithms, such as the QCancer scores, using large electronic health care databases of routinely collected primary care data to better identify at-risk patients[4,5]. These algorithms predict the absolute probability of having an undiagnosed cancer based on multiple symptoms and clinical factors, including age, sex, deprivation, lifestyle factors and clinical conditions[4–6]. National clinical guidance recommends using absolute probability estimates to

guide clinical decisions regarding further investigation and referral for patients where the positive predictive value for having cancer is above a certain threshold (e.g., 3%)[7]. As a result, validated cancer prediction algorithms have been integrated into primary care clinical computer systems to enable clinical assessment of individual patients during consultations[4,5].

By 2020, just over half of all cancers in England were diagnosed at stage 1 or stage 2[1,8]. However, further improvements are needed to achieve the 75% target by 2028[9]. Recent research has reported that changes in blood test results can predate cancer diagnoses by several years[10–12]. For example, changes in haemoglobin, white blood cell counts and platelets could represent systemic inflammation responses triggered by early stages of a clinically undetected cancer[11,13]. This raises the possibility that blood tests could usefully be incorporated into predictive algorithms, effectively acting as affordable digital biomarkers to enable earlier identification of cancer[11,14]. Such an advance

[1]Queen Mary University of London, London, UK. [2]University of Oxford, Oxford, UK. ✉e-mail: julia.hippisley-cox@qmul.ac.uk

could not only better inform clinical decisions about which patients are investigated or referred and which are reassured or reassessed but they could also improve communication about the probability of having an existing cancer with patients.

Here, we derive cancer prediction algorithms including additional predictors and affordable blood test results, using anonymised electronic health records from over 7.4 million adults in England. The algorithms predict the overall probability of having a current as yet undiagnosed cancer with the intention of improving early cancer diagnoses (rather than predicting the risk of a cancer developing over a period of time in the future). The algorithms predict both the overall probability of cancer and the separate probabilities of 15 types of cancer, including liver cancer and oral cancer, for the first time[15]. We then evaluate their performance compared with the existing cancer prediction algorithms[4,5] in two separate, large diverse populations of over 5. million people drawn from across the U.K. We used two established electronic record research databases (QResearch and CPRD Gold) containing anonymised data collected during routine clinical care, linked to hospital and mortality records.

## Results

### Study population

We included 1071 general practices in the QResearch derivation cohort, 357 in the QResearch validation cohort and 398 in the CPRD validation cohort. Supplementary Table 1 shows inclusions and exclusions for each cohort resulting in 7,464,507 people and 129,715 incident cancers in the QResearch derivation cohort; 2,637,184 people and 44,984 incident cancers in the QResearch validation cohort and 2,736,726 people and 32,328 incident cancers in the CPRD validation cohort.

Supplementary Table 2 shows the baseline characteristics of each cohort and completeness of recording of predictors included in the final model. The cohorts were broadly similar except both English cohorts had more complete data for self-reported ethnicity, smoking, alcohol and body mass index (BMI) than the CPRD cohort from the other three UK nations. Table 1 shows the numbers of men and women with each of the 15 types of cancer in each cohort. Supplementary Figure 1 and Supplementary Tables 3 and 4 show the corresponding incidence rates.

### Factors associated with increased probability of cancer

We derived two final multivariable models—Model A included clinical factors and symptoms, whilst Model B additionally included blood test results. Supplementary Table 5 shows which predictors were tested, which were included in the final models, as well as a comparison with the existing QCancer algorithms[4,5].

The values for the heuristic shrinkage[16] were all very close to one for our final models (all >0.99), indicating no evidence of over-fitting.

Supplementary Figs. 2–5 show the fully adjusted odds ratios (95% CI) for the binary and categorical predictors for each of the 15 cancer types for Model B in men and women.

Compared with the existing QCancer algorithms, we identified four additional medical conditions associated with an increased probability of cancer, three of which were associated with increased probability of liver cancer (liver cirrhosis, hepatitis B, hepatitis C). The fourth condition (HIV/AIDS) was associated with an increased probability of renal cancer in women; increased probability of colorectal, pancreatic and other cancers in men; and increased probability of blood cancers in both men and women.

We identified two additional associations with family history for lung cancer and blood cancer. We identified seven additional symptoms (itching, bruising, back pain, hoarseness, flatulence, abdominal mass, dark urine) associated with multiple cancer types, as shown in Supplementary Figs. 2–5. There were significant interactions between age and several symptoms, with stronger associations at younger ages for most cancers in men (Supplementary Fig. 6) and generally the reverse pattern in women (Supplementary Fig. 7).

Figure 1 shows the adjusted odds ratios associated with age and BMI using fractional polynomial terms for each cancer type. There were steep positive gradients associated with age for all cancer types except for testicular and cervical cancer. Decreasing BMI was associated with increased odds ratios for several cancer types in men and women whilst increasing BMI was associated with increased odds ratios for uterine and ovarian cancer in women.

Decreasing haemoglobin levels were associated with increasing odds ratios for lung, colorectal, blood and gastro-oesophageal cancers in men and colorectal, blood, gastric and liver cancers in women (Fig. 1).

## Table 1 | Numbers (%) of incident cancers in men and women in each cohort diagnosed within 2 years after study entry

| | QResearch derivation | QResearch validation | CPRD validation | QResearch derivation | QResearch validation | CPRD validation |
|---|---|---|---|---|---|---|
| | Women (col%) | Women (col%) | Women (col%) | Men (col%) | Men (col%) | Men (col%) |
| Total population | 3,622,789 | 1,277,015 | 1,373,006 | 3,841,718 | 1,360,169 | 1,363,720 |
| No cancer | 3,558,556 (98.23) | 1,254,910 (98.27) | 1,356,416 (98.79) | 3,776,236 (98.30) | 1,337,290 (98.32) | 1,347,982 (98.85) |
| Any cancer | 64,233 (1.77) | 22,105 (1.73) | 16,590 (1.21) | 65,482 (1.70) | 22,879 (1.68) | 15,738 (1.15) |
| Lung cancer | 6280 (0.17) | 2091 (0.16) | 1758 (0.13) | 7573 (0.20) | 2661 (0.20) | 1973 (0.14) |
| Colorectal cancer | 5848 (0.16) | 2003 (0.16) | 1580 (0.12) | 7438 (0.19) | 2667 (0.20) | 1959 (0.14) |
| Breast cancer | 20,464 (0.56) | 7158 (0.56) | 5727 (0.42) | n/a | n/a | n/a |
| Prostate cancer | n/a | n/a | n/a | 13,155 (0.34) | 4500 (0.33) | 3327 (0.24) |
| Blood cancer | 5116 (0.14) | 1801 (0.14) | 1430 (0.10) | 7367 (0.19) | 2593 (0.19) | 2001 (0.15) |
| Ovarian cancer | 2502 (0.07) | 872 (0.07) | 617 (0.04) | n/a | n/a | n/a |
| Renal tract cancer | 3483 (0.10) | 1190 (0.09) | 709 (0.05) | 9169 (0.24) | 3189 (0.23) | 1735 (0.13) |
| Gastro-oesophageal c | 1914 (0.05) | 683 (0.05) | 537 (0.04) | 4314 (0.11) | 1536 (0.11) | 1273 (0.09) |
| Uterine cancer | 4677 (0.13) | 1541 (0.12) | 1127 (0.08) | n/a | n/a | n/a |
| Pancreatic cancer | 1933 (0.05) | 675 (0.05) | 436 (0.03) | 2254 (0.06) | 774 (0.06) | 458 (0.03) |
| Cervical cancer | 1381 (0.04) | 459 (0.04) | 356 (0.03) | n/a | n/a | n/a |
| Oral cancer | 1094 (0.03) | 373 (0.03) | 235 (0.02) | 2465 (0.06) | 886 (0.07) | 532 (0.04) |
| Testicular cancer | n/a | n/a | n/a | 1401 (0.04) | 482 (0.04) | 300 (0.02) |
| Liver cancer | 1027 (0.03) | 364 (0.03) | 73 (0.01) | 1755 (0.05) | 601 (0.04) | 241 (0.02) |
| Other cancer | 8514 (0.24) | 2895 (0.23) | 2005 (0.15) | 8591 (0.22) | 2990 (0.22) | 1939 (0.14) |

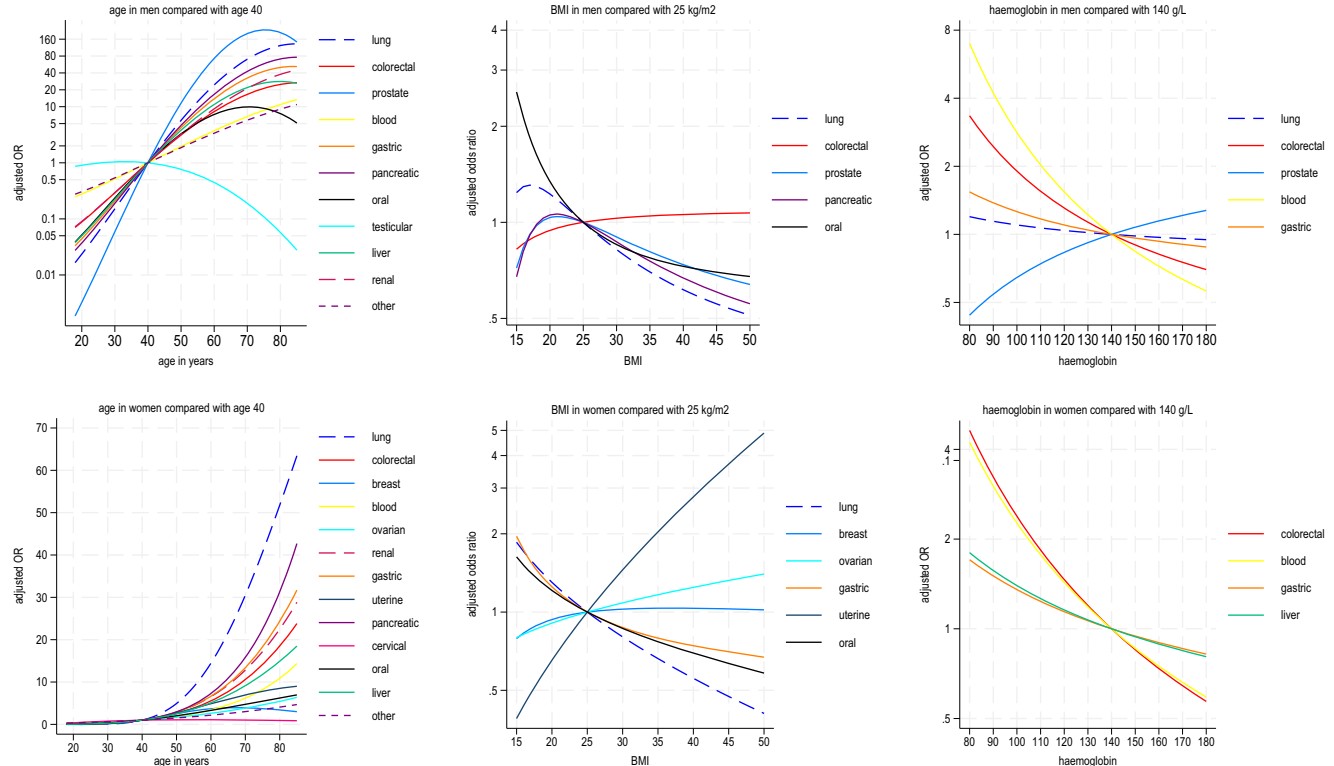

**Fig. 1 | Fractional Polynomial Terms (age, BMI, haemoglobin).** Fractional polynomial terms showing adjusted odds ratios in model B for probability of each cancer type for age, BMI and haemoglobin in men and women.

There was an inverse association between lymphocyte values and the odds ratios for most cancers except for blood cancer, where there was a strong positive association (Fig. 2).

Higher neutrophil values were associated with increasing odds ratios for most cancers in women with strongest associations for lung cancer (Fig. 2). Higher neutrophils were associated with increasing odds ratios for six cancers in men. Lower neutrophils were associated with increasing odds ratios for liver, blood and prostate cancers in men and blood cancer in women (Fig. 2).

Higher platelet values were associated with increasing odds ratios for six cancer types in men and eight in women, with strongest associations for colorectal in men and ovarian cancer in women (Fig. 2). Generally, those cancers associated with increased neutrophil counts were also associated with increased platelet counts and decreased lymphocyte counts.

Figure 3 shows the associations with liver function tests (albumin, alkaline phosphatase and bilirubin). Lower levels of albumin were generally associated with increased odds ratios for most cancers (except for prostate and renal cancer in men, where there was an inverse association). Increasing levels of alkaline phosphatase was associated with increased odds ratios for most cancers in men and women with the strongest associations for liver and pancreatic cancer. Increased bilirubin levels were strongly associated with liver cancer in men and women with similar but less strong associations for blood and pancreatic cancers.

Supplementary Table 6 summarises the directions of associations between the blood test values and the probability of different cancers in men and women.

## Discrimination
Figure 4 and Table 2 show the discrimination values using the c statistic (equivalent to the AUROC) for each cancer type for Models A and B in men and women using the QResearch validation cohort. Discrimination values tended to be higher in men compared with women and higher for Model B (with blood tests) compared with Model A (without blood tests), although the confidence intervals generally overlapped. For example, for Model B, the overall c statistic for any cancer in men was (0.876, 95% CI 0.874 to 0.878) and (0.844, 95% CI, 0.842 to 0.847) in women. The c statistic values for the specific cancer types were all above 0.78 in men. The c statistic values for 10 specific cancer types were above 0.8 in women with lower values for oral (0.747, 95% CI 0.721 to 0.774) and cervical cancer (0.694, 95% CI 0.669 to 0.719).

Supplementary Table 7 shows the number of patients in the validation cohort diagnosed at each stage of cancer. Overall, 50% of cancers in women were diagnosed at an early stage compared with 36% in men. Supplementary Table 8 shows the c statistic values for identifying patients with early stage of cancer were very similar to those for all cancer stages for both models.

The c statistic values for models A and B tended to have greater magnitude than those for QCancer, although there was some overlap of confidence intervals noting however, the results are not directly comparable due to the different age ranges (Supplementary Tables 8 and 9).

Supplementary Figs. 8–13 show discrimination values for models A and B by ethnic group, age and geographical area in the English validation cohort. Values were broadly similar within subgroups, although there was variability both between age groups (with higher values generally occurring in older patients) and by ethnic group, particularly for rarer cancers such as gastro-oesophageal, cervical and liver cancers where there were fewer events in each group.

Supplementary Table 10 shows the results for discrimination using the polytomous discrimination index (PDI) for models A and B in both validation cohorts. In women, the overall PDI for model A was 0.257 (95% CI 0.252 to 0.261) and was 0.266 (95% CI 0.261 to 0.271) for model B. For men, the PDI for men was 0.308 (0.303 to 0.314) for model A and 0.323 (0.317 to 0.329) for model B. Category-specific PDIs varied by cancer type, with the highest values (0.641) for testicular cancer in men and uterine cancer (0.439) in women for model A.

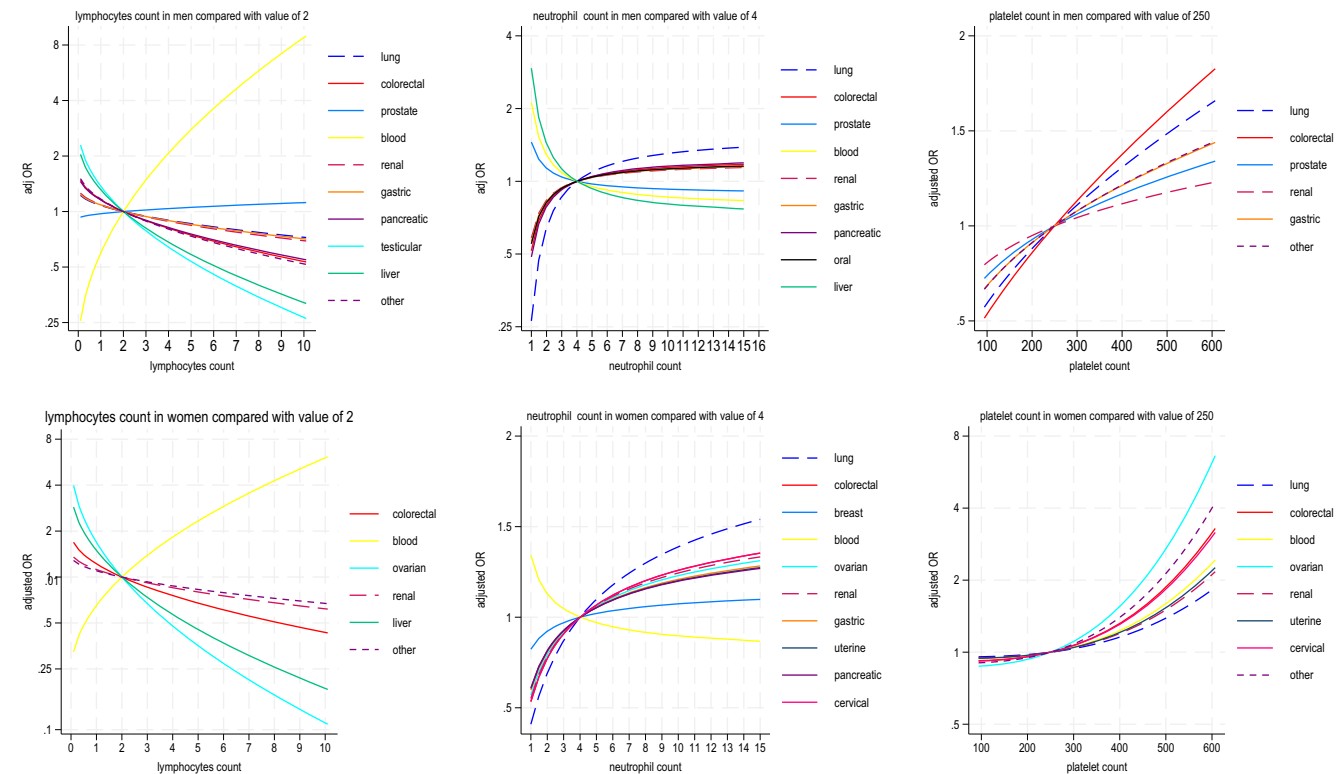

**Fig. 2 | Fractional Polynomial terms (white cells and platelets).** Fractional polynomial terms showing adjusted odds ratios for probability of each cancer type for white cell counts (lymphocytes, neutrophils) and platelets in men and women in Model B.

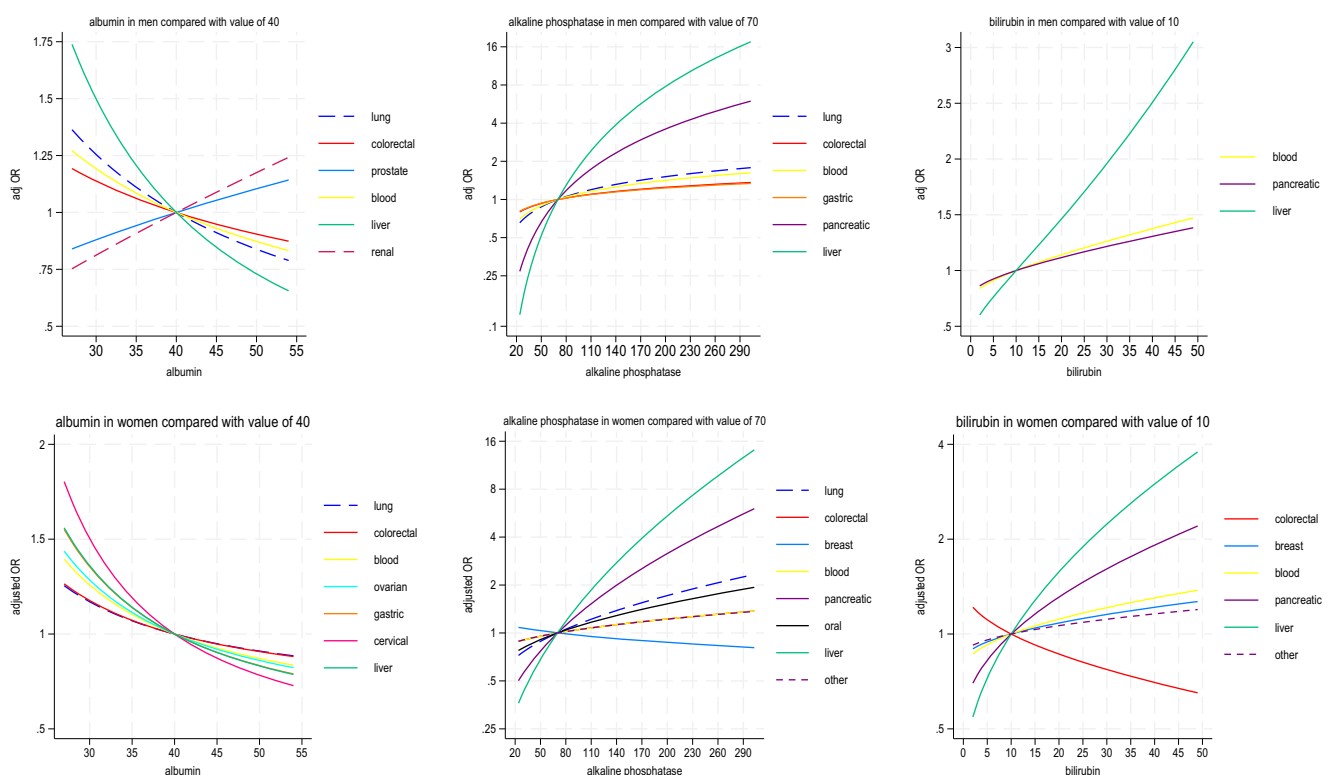

**Fig. 3 | Fractional Polynomial Terms (liver function tests).** Fractional polynomial terms showing adjusted odds ratios for probability of each cancer type for liver function tests (albumin, alkaline phosphatase and bilirubin) in men and women in model B.

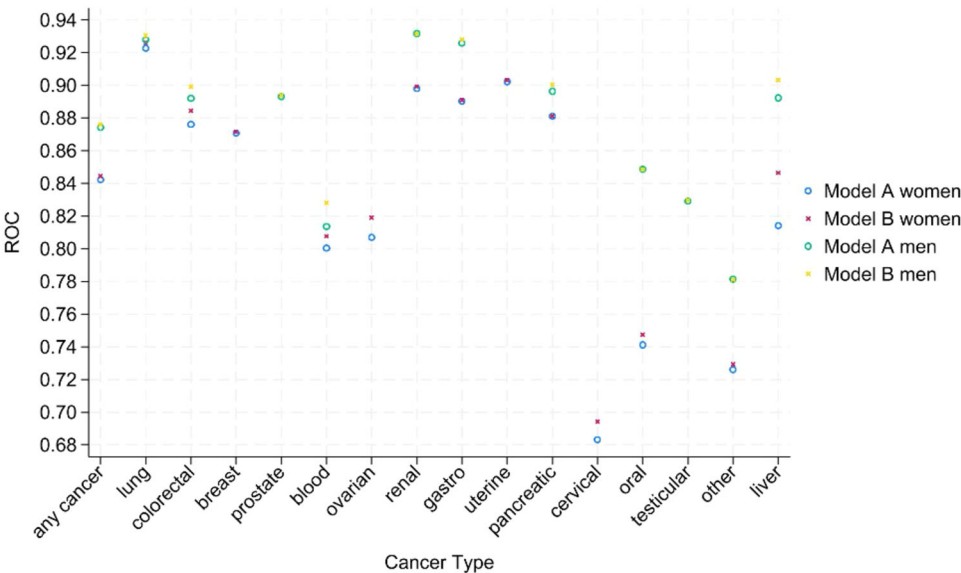

**Fig. 4 | Discrimination.** Discrimination using c statistic values for final model A (no blood test predictors) and model B (with blood tests) in men and women aged 18–84 in the QResearch validation cohort.

## Calibration

Table 3 and Supplementary Figs. 14 and 15 show that Model A and Model B were well calibrated in the England validation cohort overall and for each cancer type in men and women. For example, the calibration intercept for Model A for lung cancer in women was close to zero with a value of −0.07 (−0.11 to −0.03) with −0.02 (−0.06 to 0.02) for men. The calibration slope was 1.00 (0.97 to 1.03) for women and 0.99 (0.96 to 1.01) for men. However, there was a degree of overprediction of the probability of having cancer for all cancers in the CPRD external cohort in both men and women (Supplementary Table 8).

Supplementary Table 12 and Supplementary Figs. 16 and 17 show miscalibration of QCancer in people aged 25–84 years, with a tendency for underprediction for most cancers except for testicular and ovarian cancers.

## Decision curve analysis

The decision curves in Fig. 5 and Supplementary Figs. 18–26 indicate a very small increased net benefit of using Model B compared with Model A and a greater increase compared with the QCancer model. This means there is a similar clinical utility for Model B compared with Model A and for both models compared with a strategy of 'treating' (e.g., referring for investigation) everyone or no one[17].

## Reclassification and sensitivity

Supplementary Table 13 shows the sensitivity values using the England validation cohort for Models A, B and QCancer for identifying cancer cases at the NICE recommended referral threshold of 3%[7]. Of the 22,879 total cancer cases in men, 17,878 (78.1%) were identified using QCancer, 18,903 (82.6%) using Model A and 18,891 (82.6%) using Model B. The corresponding figures for women were 14,582 (66.0% of 22,105 cases) using QCancer, 17,110 (77.4%) using Model A and 17,024 (77.0%) using Model B. Sensitivity values varied considerably by cancer type, with model B having higher sensitivity values than model A for seven cancers (colorectal, blood, liver, lung, ovary, pancreas, prostate) and similar values for the rest.

The sensitivity of both models for identifying cancers at stages 1 and 2 was very similar to that for all cancer stages (Supplementary Table 13). For example, of the 7133 cancers diagnosed at stages 1 or 2, in women the sensitivity was 75.2% with Model A and 74.4% with model

B. For men the corresponding figures were 81.3% (Model A) and 80.9% (Model B).

Supplementary Table 14 shows characteristics of men and women reclassified using Model A compared with QCancer at the 3% threshold. There were 8311 men (0.6% of 1,360,169) who were reclassified down and 59,103 (13.4%) who were reclassified up with Model A compared with QCancer. The corresponding figures for women were 3736 (0.3%) and 87,446 (6.9%). Those reclassified up with Model A tended to be older, whilst those reclassified down tended to be younger.

## Discussion

We have developed and externally validated cancer prediction algorithms to estimate the probability of an individual having a current, as yet undiagnosed cancer, incorporating a wider range of predictors in a diverse population of men and women, all with good face validity and clinical utility. We have, for the first time, included liver and oral cancers among the 15 types of cancer predicted. We have identified several additional factors and symptoms which predict different types of cancer (for example, HIV/AIDS, which is now considered a long-term condition). Crucially, we incorporated the results of six widely available blood tests which potentially act as affordable biomarkers to improve early cancer diagnosis. We have demonstrated improved discrimination, calibration and net benefit compared with existing algorithms used in clinical practice. Overall, these algorithms had good discrimination and sensitivity both for all cancers as well as those diagnosed at an early stage (stages 1 and 2). The reclassification statistics suggest that these algorithms would result in clinically important changes in probability estimates compared with existing algorithms leading to different advice or interventions, particularly for those with additional predictors. However, whilst there was an improvement in sensitivity for Model B (with blood tests) compared with Model A (without blood tests) for seven cancers, there was little difference in discrimination and only a small improvement in net benefit.

To our knowledge, this is the only algorithm to estimate the probability of having a current as yet undiagnosed liver cancer in primary care, incorporating clinically relevant predictors including age, deprivation, smoking, type 2 diabetes, cirrhosis, hepatitis B and hepatitis C along with 12 clinically relevant symptoms and changes in full blood count and liver function tests[15]. Early diagnosis of liver

**Table 2 | Discrimination using c statistics (95% CI) by cancer type for models A and B in men and women aged 18–84 years in the QResearch English validation cohort and CPRD external validation cohort**

| | Model A women<br>C statistic (95% CI) | Model B women<br>C statistic (95% CI) | Model A men<br>C statistic (95% CI) | Model B men<br>C statistic (95% CI) |
|---|---|---|---|---|
| **QResearch** | | | | |
| Any cancer | 0.842 (0.840 to 0.845) | 0.844 (0.842 to 0.847) | 0.874 (0.872 to 0.876) | 0.876 (0.874 to 0.878) |
| Lung | 0.923 (0.917 to 0.928) | 0.926 (0.920 to 0.932) | 0.928 (0.923 to 0.932) | 0.930 (0.926 to 0.935) |
| Colorectal | 0.876 (0.869 to 0.883) | 0.884 (0.877 to 0.892) | 0.892 (0.886 to 0.898) | 0.899 (0.894 to 0.905) |
| Breast | 0.871 (0.867 to 0.875) | 0.872 (0.868 to 0.876) | n/a | n/a |
| Prostate | n/a | n/a | 0.893 (0.890 to 0.896) | 0.894 (0.891 to 0.897) |
| Blood | 0.800 (0.790 to 0.811) | 0.808 (0.797 to 0.819) | 0.814 (0.805 to 0.822) | 0.828 (0.820 to 0.837) |
| Ovarian | 0.807 (0.791 to 0.823) | 0.819 (0.801 to 0.837) | n/a | n/a |
| Renal | 0.898 (0.888 to 0.908) | 0.899 (0.889 to 0.909) | 0.932 (0.927 to 0.936) | 0.932 (0.927 to 0.936) |
| Gastro | 0.890 (0.875 to 0.905) | 0.891 (0.876 to 0.906) | 0.926 (0.919 to 0.933) | 0.928 (0.921 to 0.935) |
| Uterine | 0.902 (0.893 to 0.912) | 0.903 (0.893 to 0.913) | n/a | n/a |
| Pancreas | 0.881 (0.868 to 0.895) | 0.881 (0.866 to 0.896) | 0.896 (0.884 to 0.908) | 0.900 (0.888 to 0.913) |
| Cervical | 0.683 (0.658 to 0.709) | 0.694 (0.669 to 0.719) | n/a | n/a |
| Oral | 0.741 (0.714 to 0.768) | 0.747 (0.721 to 0.774) | 0.849 (0.835 to 0.863) | 0.849 (0.835 to 0.862) |
| Testicular | n/a | n/a | 0.829 (0.808 to 0.850) | 0.830 (0.808 to 0.851) |
| Liver | 0.814 (0.790 to 0.838) | 0.846 (0.823 to 0.870) | 0.892 (0.879 to 0.906) | 0.903 (0.889 to 0.917) |
| Other | 0.726 (0.717 to 0.735) | 0.729 (0.720 to 0.739) | 0.781 (0.774 to 0.789) | 0.781 (0.773 to 0.789) |
| | **Model A women** | **Model B women** | **Model A men** | **Model B men** |
| **CPRD** | | | | |
| Any cancer | 0.832 (0.829 to 0.835) | 0.835 (0.832 to 0.838) | 0.863 (0.861 to 0.866) | 0.865 (0.863 to 0.868) |
| Lung | 0.920 (0.915 to 0.926) | 0.923 (0.917 to 0.929) | 0.923 (0.918 to 0.928) | 0.928 (0.923 to 0.933) |
| Colorectal | 0.858 (0.849 to 0.867) | 0.873 (0.864 to 0.882) | 0.879 (0.873 to 0.886) | 0.886 (0.879 to 0.893) |
| Breast | 0.849 (0.844 to 0.854) | 0.849 (0.844 to 0.854) | n/a | n/a |
| Prostate | n/a | n/a | 0.880 (0.876 to 0.884) | 0.882 (0.878 to 0.886) |
| Blood | 0.780 (0.768 to 0.792) | 0.795 (0.782 to 0.808) | 0.816 (0.807 to 0.825) | 0.829 (0.820 to 0.838) |
| Ovarian | 0.817 (0.799 to 0.835) | 0.824 (0.805 to 0.843) | n/a | n/a |
| Renal | 0.870 (0.856 to 0.883) | 0.873 (0.860 to 0.887) | 0.916 (0.909 to 0.923) | 0.916 (0.909 to 0.924) |
| Gastro | 0.885 (0.869 to 0.901) | 0.888 (0.871 to 0.904) | 0.918 (0.911 to 0.926) | 0.921 (0.913 to 0.929) |
| Uterine | 0.906 (0.896 to 0.916) | 0.908 (0.897 to 0.919) | n/a | n/a |
| Pancreas | 0.881 (0.864 to 0.897) | 0.887 (0.869 to 0.905) | 0.886 (0.870 to 0.903) | 0.889 (0.872 to 0.906) |
| Cervical | 0.695 (0.667 to 0.723) | 0.706 (0.674 to 0.737) | n/a | n/a |
| Oral | 0.742 (0.706 to 0.778) | 0.743 (0.707 to 0.778) | 0.829 (0.810 to 0.848) | 0.830 (0.810 to 0.849) |
| Testicular | n/a | n/a | 0.773 (0.744 to 0.801) | 0.772 (0.744 to 0.801) |
| Liver | 0.788 (0.730 to 0.846) | 0.835 (0.780 to 0.889) | 0.906 (0.884 to 0.927) | 0.939 (0.920 to 0.957) |
| Other | 0.788 (0.730 to 0.846) | 0.737 (0.726 to 0.748) | 0.784 (0.775 to 0.794) | 0.786 (0.776 to 0.795) |

cancer is especially important, given the unexplained 50% rise in incidence over the last decade, the marked improvement in survival associated with a stage 1 rather than a stage 4 diagnosis and the potential for further assessment using biomarkers (such as alpha feta protein) or liver scans and introduction of new treatments such as immunotherapy[15].

Our study has good face validity since the associations are largely consistent with other studies which have investigated associations between commonly used blood tests and subsequent cancer risk[11–13,18,19]. For example, decreased levels of haemoglobin and lymphocytes and increased levels of neutrophils and platelets have been associated with an increased risk of colorectal cancer[19]. Increased platelets have been associated with increased risks of lung, prostate, lung, gastro-oesophageal and renal cancers[13,20]. Furthermore, we report four additional associations for platelets with cancer types in women (blood, ovary, cervix and uterus). Similarly, in line with our findings, low albumin has been associated with an increased probability of cancer[18]; increased levels of alkaline phosphatase have been associated with lung, liver, colorectal, gastric and pancreas cancers[21]

and increased bilirubin levels have been associated with an increased probability of pancreatic cancer in some[10] but not all studies[22], and increased probability of liver cancer[22] and a decreased probability of colorectal cancer[22]. The discrimination also improved on prediction tools designed only to predict one type of cancer[23].

Currently available QCancer algorithms are used in clinical practice in the UK, supported by national guidelines[7]. These recommend urgent referral when the probability (positive predictive value) of a patient having a current as yet undiagnosed cancer based on individual symptoms is over 3%, and hence prediction tools which can predict overall probability of a cancer have a utility in this context. The pattern of probabilities for individual cancers would help determine the most appropriate tests and investigations (and/or where to refer to). Similarly, the overall probabilities of cancer, particularly for those with non-site-specific symptoms, could help determine which patients might benefit from the non-specific cancer diagnostic pathways[7,24]. However, clinical practice needs to be based on the best possible algorithms for cancer prediction since it will materially affect which patients are offered further investigation or referrals and which are reassured or

**Table 3 | Calibration slope and calibration intercept (CITL) estimates (95% CI) by cancer type for models A and B in the QResearch English validation cohort**

|  | Model A women | Model B women | Model A men | Model B men |
|---|---|---|---|---|
| **CITL** | | | | |
| Lung | −0.07 (−0.11 to −0.03) | −0.07 (−0.12 to −0.03) | −0.02 (−0.06 to 0.02) | −0.02 (−0.06 to 0.02) |
| Colorectal | −0.04 (−0.08 to 0.01) | −0.04 (−0.08 to 0.01) | 0.00 (−0.04 to 0.04) | 0.00 (−0.04 to 0.04) |
| Breast | 0.01 (−0.05 to 0.07) | 0.01 (−0.05 to 0.07) | n/a | n/a |
| Blood | −0.01 (−0.06 to 0.03) | −0.01 (−0.06 to 0.03) | −0.01 (−0.05 to 0.03) | −0.01 (−0.05 to 0.03) |
| Ovarian | −0.01 (−0.08 to 0.06) | −0.01 (−0.07 to 0.06) | n/a | n/a |
| Renal | −0.04 (−0.10 to 0.01) | −0.04 (−0.10 to 0.01) | −0.04 (−0.08 to −0.00) | −0.04 (−0.08 to −0.00) |
| Gastric | 0.00 (−0.07 to 0.08) | 0.00 (−0.08 to 0.08) | −0.00 (−0.06 to 0.05) | −0.00 (−0.06 to 0.05) |
| Uterine | −0.11 (−0.16 to −0.05) | −0.11 (−0.16 to −0.05) | n/a | n/a |
| Pancreas | −0.02 (−0.10 to 0.06) | −0.02 (−0.09 to 0.06) | −0.04 (−0.11 to 0.03) | −0.04 (−0.12 to 0.03) |
| Cervical | −0.04 (−0.14 to 0.05) | −0.04 (−0.13 to 0.05) | n/a | n/a |
| Oral | −0.04 (−0.15 to 0.06) | −0.04 (−0.15 to 0.06) | 0.01 (−0.06 to 0.07) | 0.01 (−0.06 to 0.07) |
| Other | −0.04 (−0.08 to −0.00) | −0.04 (−0.08 to −0.00) | −0.02 (−0.06 to 0.01) | −0.02 (−0.06 to 0.02) |
| Liver | −0.01 (−0.11 to 0.10) | −0.01 (−0.11 to 0.10) | −0.05 (−0.13 to 0.03) | −0.06 (−0.14 to 0.02) |
| Prostate | n/a | n/a | −0.08 (−0.11 to −0.05) | −0.08 (−0.11 to −0.05) |
| Testicular | n/a | n/a | −0.02 (−0.11 to 0.07) | −0.02 (−0.11 to 0.07) |
| **Slope** | | | | |
| Lung | 1.00 (0.97 to 1.03) | 1.00 (0.97 to 1.03) | 0.99 (0.96 to 1.01) | 0.99 (0.96 to 1.01) |
| Colorectal | 0.99 (0.96 to 1.01) | 0.98 (0.96 to 1.01) | 1.00 (0.97 to 1.02) | 1.01 (0.98 to 1.03) |
| Breast | 1.01 (0.99 to 1.02) | 1.01 (0.99 to 1.02) | n/a | n/a |
| Blood | 0.96 (0.93 to 0.99) | 0.96 (0.93 to 0.99) | 0.98 (0.95 to 1.00) | 0.98 (0.95 to 1.00) |
| Ovarian | 1.01 (0.97 to 1.06) | 1.01 (0.97 to 1.05) | n/a | n/a |
| Renal | 1.02 (0.99 to 1.05) | 1.02 (0.99 to 1.05) | 0.99 (0.98 to 1.01) | 0.99 (0.98 to 1.01) |
| Gastric | 1.00 (0.96 to 1.03) | 1.00 (0.97 to 1.04) | 1.01 (0.98 to 1.03) | 1.00 (0.98 to 1.03) |
| Uterine | 1.00 (0.98 to 1.02) | 1.00 (0.98 to 1.02) | n/a | n/a |
| Pancreas | 1.01 (0.96 to 1.06) | 1.00 (0.96 to 1.05) | 0.99 (0.95 to 1.03) | 0.99 (0.95 to 1.03) |
| Cervical | 0.98 (0.92 to 1.04) | 0.98 (0.92 to 1.04) | n/a | n/a |
| Oral | 0.97 (0.91 to 1.02) | 0.97 (0.91 to 1.02) | 0.98 (0.95 to 1.01) | 0.98 (0.95 to 1.01) |
| Other | 0.99 (0.95 to 1.03) | 0.99 (0.95 to 1.03) | 0.99 (0.96 to 1.03) | 0.98 (0.95 to 1.02) |
| Liver | 0.99 (0.92 to 1.05) | 0.99 (0.94 to 1.04) | 0.97 (0.92 to 1.01) | 0.96 (0.92 to 0.99) |
| Prostate | n/a | n/a | 1.00 (0.97 to 1.03) | 0.99 (0.96 to 1.02) |
| Testicular | n/a | n/a | 1.02 (0.98 to 1.05) | 1.02 (0.98 to 1.06) |

monitored. Failure to adequately assess the probability of a patient having cancer and offer appropriate investigations or interventions across all relevant patient groups could further disadvantage vulnerable patients, particularly those with significant long-term conditions which are now incorporated (hepatitis B and C, HIV/AIDS and liver cirrhosis). Whilst these algorithms do not result in a clinical diagnosis of cancer, the identification of high-risk people could lead to targeted interventions to expedite cancer diagnosis and earlier treatment, which in turn could then improve cancer survival. It is indeed likely that some of the cases would have been diagnosed shortly already, but these models can identify high-risk patients based on combinations of clinical factors and may particularly be helpful for some more rare cancers which present less frequently in primary care where there can be less awareness of red flag symptoms. As stated above the models are not intended to be used for screening for a cancer which might develop in the future over a longer period of time (such as 5 or 10 years).

The strengths and limitations of this study are similar to those of other well-established prediction tools[4,5,25,26]. The strengths include clinically relevant and explainable predictors, large study size, representativeness, lack of selection, recall and respondent bias, and no evidence of over-fitting[3]. The inclusion of more granular information on predictors is a strength, in that the predictions for individual patients are likely to better reflect their individual probability of having cancer, although this needs to be balanced against the increased complexity of implementing the algorithm. However, this is mitigated in settings where electronic health records are available since most relevant information is already available at the point of care and can be automatically populated[27]. Whilst we report improved discrimination for the two models compared with existing tools, the absolute values of the improvement in the c statistics were small, although this needs to be interpreted in the context of other relevant measures including calibration and reclassification[28]. Our study has good face validity since it was conducted in a setting where most patients are managed, and hence the models could be implemented in similar clinical settings, subject to local validation or recalibration.

Limitations include the lack of formal adjudication of cancer diagnoses. However, the use of linked hospital, mortality and cancer registry data ensures good ascertainment of cancer outcomes in the English cohorts. The over-prediction observed in the other three U.K. nations is likely to reflect the lack of available linked cancer outcome data from secondary care in these nations in the CPRD cohort. Recording of symptoms in primary care records may be less complete or accurate than for diagnostic codes since patients might not visit their GP with mild symptoms, may not report all symptoms to their GP when they do consult, or GPs might not record all the symptoms in the

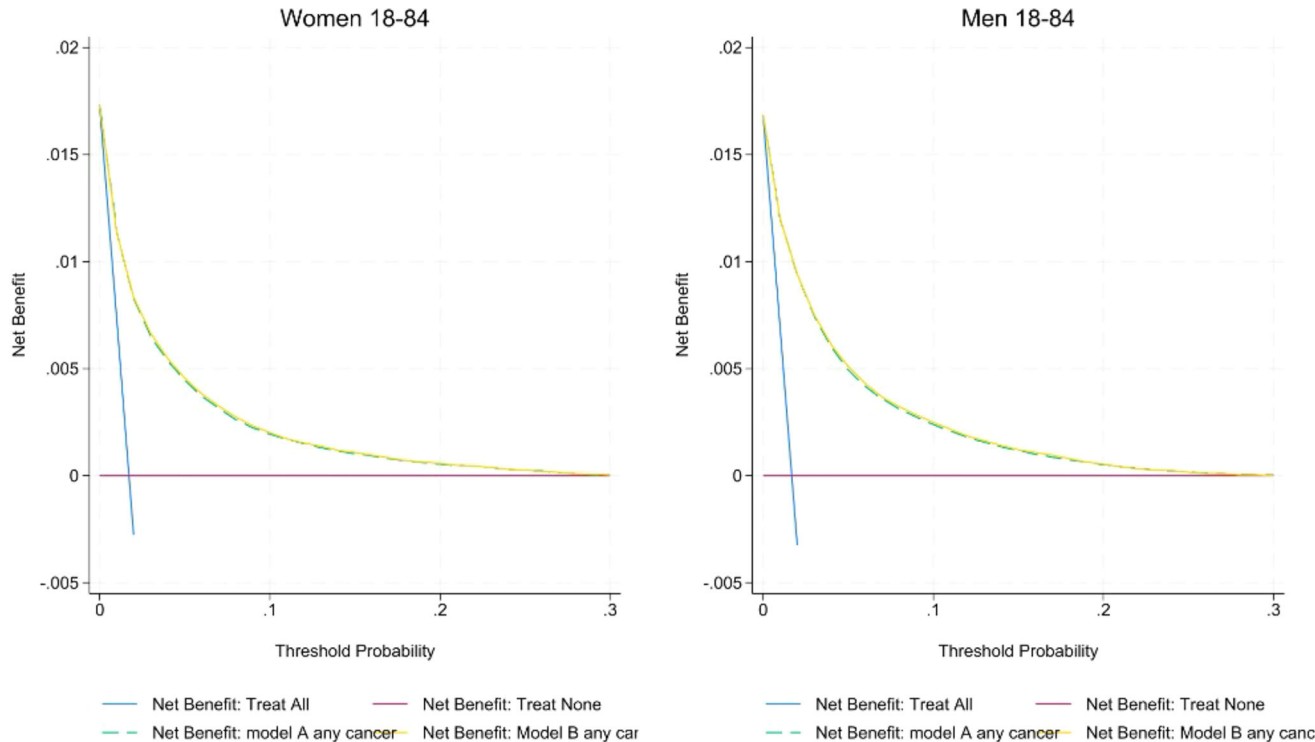

**Fig. 5 | Decision Curve Analysis.** Decision curve analysis comparing probability of any cancer using model A and Model B in men and women aged 18–84 in the English validation cohort.

electronic health record. The effect of this information or recording bias could be to underestimate odds ratios if symptoms are not reported or not recorded, or to over-inflate the odds ratios if only the more severe symptoms were reported and recorded. Similarly, family history of some types of cancer might be under-recorded since it is not routinely assessed and recorded in GP records. Whilst there is potential for bias due to missing data for BMI, smoking and alcohol, our data are substantially more complete than previous studies with residual biases mitigated by multiple imputation. We expect that missing data during a consultation using a cancer prediction tool will be collected from the patient/carer by their clinician, so missing data for the implementation of these models is unlikely to be a significant issue. Whilst our validation covers a fully external population, further research should validate the models in different countries outside the UK with different cancer rates.

We have developed two diagnostic prediction models to estimate the probabilities of patients having an existing, as yet undiagnosed cancer. They predict overall cancer probability as well as the probabilities of 15 individual types of cancer. Both models are based on simple clinical variables, which are already stored in patient electronic health records or can be ascertained in clinical practice, including age, sex, medical conditions, lifestyle factors, and symptoms. Model B additionally includes blood test results where these are available. Whilst the models themselves do not make a diagnosis of cancer, they can be used to triage patients into those with the highest probabilities of having cancer who need further assessment or referral, potentially making the diagnostic pathway more efficient. Both models performed well in identifying patients with the highest probabilities of cancer in an external validation cohort with good discrimination and calibration, and they both improve on the existing QCancer models which are used in the NHS[4,5].

## Methods

This research complies with all relevant ethical regulations. The QResearch ethics approval by the East Midlands-Derby Research Ethics Committee [reference 18/EM/0400]. The CPRD ERAP approval reference is 20_000162.

### Study design and data sources

We undertook community-based cohort studies using two large electronic medical records databases, QResearch (version 48) and the Clinical Practice Research Datalink (CPRD Gold). We randomly allocated three-quarters of QResearch practices in England to the derivation cohort and the remainder to an internal English validation dataset. Both CPRD and QResearch are based on anonymised medical record data collected during the course of clinical care. QResearch is based on a commercial computer system known as EMIS (Egton Medical Information Systems) whilst CPRD (Gold) is based on a different commercial system (Vision). We used CPRD practices from three U.K. nations (i.e., Northern Ireland, Scotland and Wales) to create a second fully external geographically distinct validation cohort.

We included people aged 18–84 years registered between January 1, 2015, and March 31, 2023. We included patients from age 18 years to incorporate cancers that can affect younger people such as haematological malignancies, breast, cervical and testicular cancers.

Entry to the cohort was the latest of the following: study start date (1st January 2015), 12 months after the patient registered with the practice, date of 18th birthday, and for those patients with one or more 'red flag symptoms', the date of first recorded consultation with a symptom within the study period. Where patients had onset of multiple red flag symptoms recorded, the entry date was the earliest recorded date of a red flag symptom in the study period. We excluded patients with a recorded red flag symptom in the 12 months prior to the cohort entry date. We followed patients up until the earliest date of the diagnosis of cancer, death, de-registration with the practice, 2 years from study entry or the study end date (31st March 2023). Patients with less than 2 years of follow-up were excluded from the analyses unless they had a cancer diagnosis or died within 2 years.

## Outcome definitions

Our primary outcome for model development was a diagnosis of cancer (excluding non-melanoma skin cancers) occurring within 2 years after study entry recorded on any of the four linked data sources (GP, hospital, mortality and cancer registry). A 2-year time horizon was used to identify patients with a current as yet undiagnosed cancer at study entry in line with other studies[4,5]. We used the earliest date of diagnosis from any data source as the index date. We identified 15 cancer types using the relevant diagnostic codes (Supplementary Table 15). We included 13 types already included in QCancer (lung, colorectal, prostate, breast, blood, ovarian, renal tract, gastro-oesophageal, uterine, pancreatic, testicular, cervical and other cancer) plus oral-pharyngeal and liver cancer since there was a sufficient number of diagnoses for analysis.

For model validation, we used the above outcome definition for the QResearch validation cohort. For CPRD, the definition was based solely on cancer diagnoses recorded in the GP records since linked data for deaths, cancer registry and hospital admissions were unavailable.

## Predictor variables

We included candidate predictors based on existing algorithms and the literature[4,5] as shown in Supplementary Table 5 with separate models by self-reported sex using the self-report information included in the NHS record. Predictors include factors which contribute to baseline probability of cancer, such as demographic factors, smoking, alcohol, family history of cancer, comorbidities, as well as factors which could indicate a suspected but as yet undiagnosed cancer (e.g., blood test results and symptoms). Symptoms were considered into two broad groups (1) 'red flag' symptoms, which have been identified in the literature as being associated with cancer[4,5,29] or clinical guidance for urgent referral[7] and (2) other less specific symptoms, which may nonetheless indicate an increased probability of having cancer[4,5]. We included commonly used blood tests recorded in the 2 years prior to cohort entry.

## Model development

We used multinomial logistic regression to estimate the coefficients for each predictor variable for each type of cancer. We fitted separate models for men and women. We used 11 types of cancer in men and 13 types in women, a category for "other cancer types" and a category for "no cancer" as the categorical outcome variable. We restricted the analyses to patients who had a cancer diagnosis within 2 years of cohort entry or had at least 2 years of follow-up since this represents the period of time during which existing cancers are likely to become clinically manifest[29,30].

We used multiple imputation with chained equations to replace missing values for categorical variables (alcohol, smoking)[31] and continuous variables (body mass index, haemoglobin, platelets, neutrophils, lymphocytes, albumin, bilirubin, alkaline phosphatase, alanine transaminase, random glucose and HBA1C) which were log-transformed. We carried out five imputations separately for men and women. We used Rubin's rules to combine the results across the imputed datasets[32]. For binary variables, we coded them as present if there was a recorded diagnosis in the GP record and otherwise coded them as absent.

We fitted full models initially and retained variables in the overall model that were significant at the 0.01 level. We constrained coefficients to equal zero for individual types of cancer within the overall model where the risk ratio was between 0.80 and 1.20 (for binary variables) and was not significant at the 0.01 level. We used *P*-values along with effect sizes (odds ratios) to develop parsimonious models. We have previously considered automated variable selection approaches but prefer this approach because it is not based solely on statistical significance, and some automated approaches have not been extended to multinomial logistic models.

We used fractional polynomials to model non-linear relationships with continuous variables[33]. We tested for significant interactions between predictor variables and age. We assessed model optimism by calculating heuristic shrinkage[16]. We derived two final models: a simpler Model A including predictors and symptoms and Model B additionally incorporating blood test results.

We combined the regression coefficients for each predictor variable from the final models with the constant terms to derive equations to estimate the probabilities of having each type of cancer. We estimated the absolute probability of an individual having any type of cancer by summing the probabilities across the individual cancer types as in previous studies[4,5].

## Model evaluation

We used multiple imputation in both validation cohorts to replace missing values for categorical variables (alcohol, smoking) and continuous variables (which were log-transformed), separately for men and women. We applied the prediction equations for our final models A and B to estimate absolute probabilities for each type of cancer. We evaluated performance in both validation cohorts overall and by geographical area, self-reported ethnicity and age group where there were at least five cancer cases in each subgroup.

We assessed discrimination by calculating the c statistic, which is equivalent to the Area under the Receiver Operating Curve (AUROC) statistic for each cancer type, combining the results using Rubin's rules. We used single imputed datasets for other performance statistics. We also calculated the polytomous discrimination index (PDI) for the model overall and each cancer type[34,35]. The PDI is a summary measure of the simultaneous discrimination between all outcome categories (12 categories for men and 14 for women including no cancer). For an outcome variable with $n$ categories, the PDI will equal $1/n$ for discrimination no better than chance and a maximum value of 1 indicates perfect discrimination between categories.

We assessed calibration by plotting the mean predicted probability of having cancer with the observed probability by centiles of predicted probabilities. We also calculated the calibration slope and intercept[36]. The calibration slope indicates whether the probability predictions are too extreme (if <1) or too narrow (if >1). The calibration intercept indicates whether the probabilities are systematically overestimated (if <0) or underestimated (if >0)[36].

In order to determine the performance of the algorithms to predict cancers at an early stage, we undertook additional analyses using the QResearch validation cohort which were limited to cancers diagnosed during the period 2015–2020 to match the period during which cancer stage at diagnosis was available on the linked cancer registry data. We defined early-stage cancer as cancers diagnosed at stages 1 or 2 in line with national guidance.

## Decision curve analysis

We used decision curve analysis in both validation cohorts to evaluate the net benefits of each model in people aged 25–84 years (to match the applicable age range for QCancer), compared with alternative strategies such as assuming all people were treated or no one was treated. Decision curve analysis calculates a clinical 'net benefit' for one or more prediction models in comparison to default strategies of 'treating' (e.g., referring for investigation) all or no patients. It is calculated over a range of threshold probabilities[17]. The strategy with the highest net benefit at any given threshold is considered to have the most clinical value[37].

## Stratification statistics

We used the validation cohorts to calculate sensitivity and specificity values using a threshold for the probability of having any cancer and for each cancer type, restricting the analyses to those who had any cancer within 2 years or had at least 2 years of follow-up. We classified

individuals as "high risk" of having cancer if their predicted probability was ≥3%, in line with the threshold in current U.K. guidelines[7]. We calculated predicted probabilities to determine the percentage of people reclassified by the models at this threshold both for all cancer diagnoses and those diagnosed at stages 1 and 2.

## Sample size calculations

We used all eligible individuals to develop and validate the models to maximise the power and generalisability of the results. With the sample sizes in the derivation cohorts for men and women, we had large enough samples to consider up to 100 predictor variables and target a shrinkage value of 0.9, a difference of 0.05 for comparison of $R^2$ values and a margin of error of 0.05 for the estimates of overall probability, with the exception of a small number of more rare cancer pairs where a reduced shrinkage of 0.8 can be targeted with a smaller number of predictor variables considered for inclusion.

We performed sample size calculations for both model derivation[38] and validation[39], as shown in the Supplement. We used Stata (version 18) for analyses.

## Reporting summary

Further information on research design is available in the Nature Portfolio Reporting Summary linked to this article.

## Data availability

The electronic health data generated in this study are derived from the QResearch and CPRD database under accession code OX303 https://www.qresearch.org/research/research-programs-and-projects/comparisons-of-risk-prediction-algorithms-using-three-clinical-research-databases-qresearch-cprd-aurum-and-cprd-gold/. The raw data are not available under restricted access for to ensure compliance with patient privacy, ethical and license conditions for access to patient electronic health records in England. The processed data access can be obtained by following the published information on the QResearch website (www.qresearch.org) and CPRD website (www.cprd.com).

## Code availability

Clinical codes are published under a Creative Commons licence here: https://www.qresearch.org/data/qcode-group-library/ with accompanying details in Supplementary Table 1. The software which implements the algorithm is available here: https://canpredict.org/ED/academic-use-source-code/.

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

## Acknowledgements

We acknowledge the contribution of EMIS practices who contribute to QResearch and EMIS Health. This project involves data derived from patient-level information collected by the NHS, as part of the care and support of cancer patients. The hospital, cancer and mortality data are collated, maintained and quality assured by the National Disease Registration Service which is part of NHS England. Access to the data was facilitated by the NHS England Data Access Request service. NHS England bears no responsibility for the analysis or interpretation of the data. This study was funded by the John Fell Fund, University of Oxford.

## Author contributions

Julia Hippisley-Cox initiated the study, development of the research question, undertook the literature review, data extraction, data manipulation and primary data analysis and wrote the first draft of the paper. Carol Coupland contributed to the refinement of the research question, design, analysis, interpretation and drafting of the paper. These authors jointly supervised the work and approved the final submission.

## Competing interests

All authors have completed the ICMJE uniform disclosure form at www.icmje.org/coi_disclosure.pdf and declare: J.H.C. reports grants from National Institute for Health Research, John Fell Oxford University Press Research Fund, Cancer Research U.K. (C5255/A18085), and other research councils, during the conduct of the study. J.H.C. is an unpaid director of QResearch, a not-for-profit organisation which is a partnership between the University of Oxford and EMIS Health who supply the QResearch database used for this work. Until 9 Aug 2023, J.H.C. had a 50% shareholding in ClinRisk Ltd, co-owning it with her husband, who was an executive director. On 9 August 2023, 100% of the share capital was donated to Endeavour Health Care Charitable Trust and the company was renamed to Endeavour Predict Ltd. J.H.C. is a consultant to Endeavour Predict Ltd. and her husband is a non-executive director to cover the transition. The company licences software both to the private sector and to NHS bodies or bodies that provide services to the NHS (through GP electronic health record providers, pharmacies, hospital providers and other NHS providers). This software implements algorithms (including QRISK3) developed from access to the QResearch database during her time at the University of Nottingham. C.C. reports receiving personal fees from ClinRisk Ltd., outside this work.
