## [Transparent Peer Review file · Nature Communications]

Development and external validation of new prediction algorithms to improve early diagnosis of cancer

Corresponding Author: Professor Julia Hippisley-Cox

Version 0:

Reviewer comments:

Reviewer #1

(Remarks to the Author)

This is a well written study on risk prediction models for several cancer based on UK electronic health records. The authors develop models for several cancers based on their previous work on Q cancer algorithms. I have several comments:

- I could not find the time horizon for the prediction and this is of paramount importance. Are these models predicting imminent diagnoses or long term risk? Risk models without a time horizon do not mean much in terms of clinical decisions. I suspect these models have very high AUC values as they predict imminent cancer diagnosis, probably cancers that are already established. Can you provide models with different time horizons, eg 2 years, 5 years and >10 years? This will have different implications for screening. Are the models with very high AUC values diagnosing rather than predicting well established cancers maybe at a progressed stage? What clinical implications do they have?

- It is unclear to me if these models produce risk probabilities of several cancers all together? How is this information going to be used to achieve impact? Is increased risks of several cancer going to be followed up with screening for different cancers?

- For the construction of the models, the authors use predefined variables which are kept in the model based on P values. Did you consider using variable selection approaches and a longer list of variables?

- Missing data in these records has been shown not to be missing at random. The selected imputation methods do not deal with this and more thorough reporting is needed to see the N of missing values and the potential bias associated with missingness

- The authors acknowledge the fact that primary care coding of cancer may miss some cancers. CPRD has linkage to HER and cancer registries, have you tried to use this information to supplement diagnoses? Liver cancer seems to have a much lower rate in CPRD compared to the Qreserach.

- The coefficients and the variables contributing in each model need to be clearly provided to allow further validation.

- The authors state in the discussion that this information can be used to triage patients into those at highest risk who need further assessment or referral but do not provide convincing examples how this can be used for the N cancers examined here. It is likely that the models at least for some cancers identify cases which would have been diagnosed shortly already in a progressed stage. There is no information on whether the model can identify early signs or long term risk. Shorter term risk would be important for cancers with screening programmes.

Reviewer #2

(Remarks to the Author)

This manuscript is well-written and clear. It describes the development of multi-variable models to predict the probability of a cancer diagnosis within two years of presentation.

Major comment

1. I think there needs to be greater clarity about the underlying purpose of the model. The lack of clarity is, I think, caused by imprecise use of the term risk, which means the probability of a future event (that has not yet occurred). A measure of risk would normally have time in the denominator. There is a fundamental difference between i) a model that uses multiple variables to indicate the probability that a cancer is already present, i.e. given this set of symptoms, signs and test results, what is the probability that this person has cancer, and ii) a model that uses multiple variables to estimate the probability that a person who does not have cancer will develop cancer at some time in the future (in a given time period). The first of these is what is relevant when considering algorithms for earlier diagnosis, but it is not a risk, it is a probability. The second of these is a risk. These two concepts are somewhat conflated in these analyses because the probability of being diagnosed with cancer in the next two years is being estimated. I think the premise is that a cancer diagnosis soon after the initial observations is being used as a proxy for the presence of a cancer at the time of presentation. Indeed on line 261 the phrase "to predict risk of a current as yet undiagnosed liver cancer" is used – thus using the term risk inaccurately as what is meant is the probability of an undiagnosed cancer. If so, this need to be made much clearer (and the term cancer risk needs to be changed to risk of a cancer diagnosis). If the aim is really to estimate future risk of incident disease (in someone who does not have cancer) then they cannot make any claims about earlier diagnosis.

2. Two prediction models were derived. Model A included only clinical variables and Model B included clinical variables with blood test results. The addition of the blood test results makes almost no difference to the performance of the model. This needs to be described more clearly and the clinical implications discussed.

3. A huge number of prediction model performance parameters are presented and it is rather difficult to see the wood for the trees. It seems to me that the key parameters are cancer site-specific sensitivity and specificity at a given cancer probability threshold. I.e. what is the performance of the model under typical scenarios in which it might be used. A cancer probability of 3% or greater is deemed clinically relevant by NICE and I think Supp Table 10 is the results table that is of major interest clinically. The overall sensitivity for detecting any cancer is not of much interest because what is needed is the probability of a specific cancer so appropriate investigations can be carried out in order to make the diagnosis.

Minor comments

4. Line 205 and following. The text should indicate the overall calibration (predicted numbers of cancer compared to observed) with some description of how that varied by specific cancer. A summary of the other key calibration parameters of interest (calibration intercept and calibration slope) should also be reported in the main text, not just the tables and figures. This should include some interpretation of the various parameters.

5. Line 217. Decision curve analysis is not commonly used in the reporting of prediction models and I think a more detailed explanation of what the numbers mean and how to interpret them would be helpful. Given that the management decisions are cancer site specific this part of the paper needs to be cancer site specific. A decision curve analysis related to any cancer makes little sense to me.

6. Line 240. Again the main text needs some summary numbers rather than simply referencing a supplementary table. It ought to be possible to read and understand the key points of a manuscript without referring to the supplement at all.

7. Line 436. The methods for combining the various validation parameters across the multiple imputed data sets needs to be described.

8. Extended figure 5. The description of how these curves were derived need to be clearer. If they are based on the regression of the mean predicted risk against observed within risk centiles then the individual points need to be plotted (as standard calibration plots) (may also include 95% confidence intervals) in order to demonstrate the actual calibration at different risk estimates.

Version 1:

Reviewer comments:

Reviewer #1

(Remarks to the Author)

I wish to thank the authors for their reply to our comments.

I still believe the clarity on the purpose of the model and the risk vs probability is still unresolved and the impact of the overall probability of cancer unclear.

Reviewer #2

(Remarks to the Author)

The authors have chosen not to change the language in which the term risk is used to mean a probability without any notion of time in the denominator. They cite others who have used this term in the same way. The fact that others have adopted what in my view is a poor practice is no defence. I think that greater specificity in the use of scientific terms aids understanding. The first comment of reviewer #1 is a perfect illustration of the confusion caused by the language used ("I could not find the time horizon for the prediction and this is of paramount importance."). This confusion would not have arisen if the term risk was not used to mean a simple probability. The very first sentence of the abstract "Cancer risk prediction algorithms are used in the UK to identify individuals at high risk of having a current, as yet undiagnosed cancer" illustrates the issue nicely as "Cancer risk prediction algorithms are also used in the UK to identify cancer-free individuals at high risk of developing cancer in the future". The changes required in this manuscript would be small. However, I accept that my opinion is no gold standard.

I agree with the reviewer that the coefficients ought to be provided. I do not think it is acceptable to hide code behind an academic license. Especially code based on data generated using publicly funded resources.

In other respects, the authors have addressed the my comments and those of the other reviewer.

Version 2:

Reviewer comments:

Reviewer #1

(Remarks to the Author)

No other comments, thank you for your responses.

(Remarks on code availability)

This is quite complex to review and my impression is that is hard to use again. Also, will it be available for everyone?

Reviewer #2

(Remarks to the Author)

All my comments have been addressed.

(Remarks on code availability)

I tried logging in to review the code, but the login failed repeatedly.

I stand by my comments that a table of coefficients should be part of the paper. The advice of a technology transfer office (Oxford Innovations) does not support good academic practice.

Open Access This Peer Review File is licensed under a Creative Commons Attribution 4.0 International License, which permits use, sharing, adaptation, distribution and reproduction in any medium or format, as long as you give appropriate credit to the original author(s) and the source, provide a link to the Creative Commons license, and indicate if changes were

made.

Manuscript: NCOMMS-24-30307-T "Development and external validation of new cancer risk algorithms to improve early diagnosis of cancer in 12.8 million people in primary care"

AUTHORS RESPONSE TO REVIEWER COMMENTS 13.08.2024

Reviewer #1 (Remarks to the Author): expertise in public health and GP healthcare data

This is a well written study on risk prediction models for several cancer based on UK electronic health records. The authors develop models for several cancers based on their previous work on Q cancer algorithms. I have several comments:

- I could not find the time horizon for the prediction and this is of paramount importance. Are these models predicting imminent diagnoses or long-term risk? Risk models without a time horizon do not mean much in terms of clinical decisions. I suspect these models have very high AUC values as they predict imminent cancer diagnosis, probably cancers that are already established. Can you provide models with different time horizons, eg 2 years, 5 years and >10 years? This will have different implications for screening. Are the models with very high AUC values diagnosing rather than predicting well established cancers maybe at a progressed stage? What clinical implications do they have?

RESPONSE: We have updated the introduction which now includes the following sentence on page 3 *"The algorithms predict overall risk of current as yet undiagnosed cancer with the intention of improving early cancer diagnoses (rather than to predict the risk of a cancer developing in the future)."*

We have updated the outcome definition in the methods which now states *"Our primary outcome for model development was a diagnosis of cancer (excluding non-melanoma skin cancers) occurring within two years after study entry recorded on any of the four linked data sources (GP, hospital, mortality and cancer registry). A two-year time horizon was used to identify patients with a current as yet undiagnosed cancer in line with other studies^{1,2}." We have separately published models for longer time periods (e.g. 5 or 10 years) which are intended for screening^{3,4}.*

The discussion already included a relevant sentence as follows *"We have developed and externally validated new cancer risk algorithms to predict risk of a current, as yet undiagnosed cancer, incorporating novel predictors in a diverse population of men and women, all with good face validity and clinical utility." And "The algorithms had good discrimination and sensitivity both for all cancers as well as those diagnosed at an early stage (stages 1 and 2). The reclassification statistics suggest that the new algorithms would result in clinically important changes in risk leading to different advice or interventions, particularly for those with the newly identified predictors"*

It is not intended that these models are used for screening, rather that they are used to identify high risk patients for referral for diagnostic tests and investigations to help promote earlier cancer diagnosis.

- It is unclear to me if these models produce risk probabilities of several cancers all together? How is this information going to be used to achieve impact? Is increased risks of several cancer going to be followed up with screening for different cancers?

RESPONSE: The intended use of the models is to support earlier cancer diagnosis (similar to existing QCancer models) rather than screening as noted above. The risk models produce risks of having individual types of cancer and then these risks are added to give an overall risk of having any type of cancer. We have updated the discussion to include the following text. *“Currently available QCancer risk algorithms are used in clinical practice in the UK, supported by national guidelines⁵. These recommend urgent referral when the overall risk of a current as yet undiagnosed cancer is over 3%. The pattern of risks for individual cancers would help determine the most appropriate tests and investigations (and/or where to refer to)”*

- For the construction of the models, the authors use predefined variables which are kept in the model based on P values. Did you consider using variable selection approaches and a longer list of variables?

RESPONSE: We list the variables considered in Extended Table 3 as explained in the methods. These were candidate predictors based on existing algorithms and reported in the literature and we did not consider any additional variables. We have added the following text to page 11 of the methods *“We used P values along with effect sizes (risk ratios) to develop parsimonious models. We have previously considered automated variable selection approaches but prefer this approach because it is not based solely on statistical significance, and some automated approaches have not been extended to multinomial logistic models”*.

- Missing data in these records has been shown not be missing at random. The selected imputation methods does not deal with this and more thorough reporting is needed to see the N of missing values and the potential bias associated with missingness

RESPONSE: as noted in the results, Extended table 1 shows the completeness of recording of predictors in the final model. We have also included a Supplementary table 13 comparing characteristics of those with complete and incomplete data. Patients with complete data tended to be older, more likely to be female, of white ethnicity.

- The authors acknowledge the fact that primary care coding of cancer may miss some chances. CPRD has linkage to HER and cancer registries, have you tried to use this information to supplement diagnoses? Liver cancer seems to have a much lower rate in CPRD compared to the QResearch.

RESPONSE: Unfortunately, linked data is not available for CPRD Gold in Scotland, Wales and Northern Ireland so this was not possible. We have updated the following sentence in the discussion which now reads *“The over prediction observed in the other three U.K. nations is likely to reflect the lack of available linked cancer outcome data from secondary care in these nations in the CPRD cohort”*.

- The coefficients and the variables contributing to each model need to be clearly provided to allow further validation.

RESPONSE: Software implementing the model with the functional form, coefficients etc will be available under an academic license from Oxford University Innovations. We have added this to Code Availability section on page 14.

- the authors state in the discussion that this information can be used to triage patients into those at highest risk who need further assessment or referral but do not provide convincing examples how they can be used for the N cancers examined here. It is likely that the models at least for some cancers identify cases which would have been diagnoses shortly already in

a progressed stage. There is no information on whether the model can identify early signs or long-term risk. Shorter term risk would be important for cancers with screening programmes.

RESPONSE: We have added some examples to the results section. We have also added more emphasis to the analyses we undertook by stage to address the points made. We have added the following text to the discussion: *“As stated above the models are not intended to be used for screening for a cancer which might develop in the future. It is indeed likely that some of the cases would have been diagnosed shortly already, but these models can identify high risk patients based on combinations of risk factors and may particularly be helpful for some more rare cancers which present less frequently in primary care where there can be less awareness of red flag symptoms”*.

Reviewer #2 (Remarks to the Author): expertise in epidemiology risk model development

This manuscript is well-written and clear. It describes the development of multi-variable models to predict the probability of a cancer diagnosis within two years of presentation.

RESPONSE: Many thanks for this summary.

Major comment

1. I think there needs to be greater clarity about the underlying purpose of the model. The lack of clarity is, I think, caused by imprecise use of the term risk, which means the probability of a future event (that has not yet occurred). A measure of risk would normally have time in the denominator. There is a fundamental difference between i) a model that uses multiple variables to indicate the probability that a cancer is already present, i.e. given this set of symptoms, signs and test results, what is the probability that this person has cancer, and ii) a model that uses multiple variables to estimate the probability that a person who does not have cancer will develop cancer at some time in the future (in a given time period). The first of these is what is relevant when considering algorithms for earlier diagnosis, but it is not a risk, it is a probability. The second of these is a risk. These two concepts are somewhat conflated in these analyses because the probability of being diagnosed with cancer in the next two years is being estimated. I think the premise is that a cancer diagnosis soon after the initial observations is being used as a proxy for the presence of a cancer at the time of presentation. Indeed, on line 261 the phrase “to predict risk of a current as yet undiagnosed liver cancer” is used – thus using the term risk inaccurately as what is meant is the probability of an undiagnosed cancer. If so, this need to be made much clearer (and the term cancer risk needs to be changed to risk of a cancer diagnosis). If the aim is really to estimate future risk of incident disease (in someone who does not have cancer) then they cannot make any claims about earlier diagnosis.

RESPONSE: We have considered this point and prefer to keep to the reference to risk as we have now clarified the outcome and purpose. There is consistent with a recent paper⁶ which states *“Diagnostic prediction models aim to calculate an individual's risk that a disease is already present”*, and *“A clinical prediction model is usually developed to estimate probabilities, often simply called risk, for individual patients with the aim to inform them, their relatives and healthcare providers about diagnosis or prognosis, to help with making better (shared) testing and treatment decisions, or for making risk-stratifications for therapeutic*

trials". That said, we have updated the first sentence of the discussion which now reads "We have developed and externally validated new cancer risk algorithms to predict the risk (probability) of an individual having a current, as yet undiagnosed cancer, incorporating novel predictors in a diverse population of men and women, all with good face validity and clinical utility".

2. Two prediction models were derived. Model A included only clinical variables and Model B included clinical variables with blood test results. The addition of the blood test results makes almost no difference to the performance of the model. This needs to be described more clearly and the clinical implications discussed.

RESPONSE: For discrimination we have updated the results as follows "Discrimination values tended to be higher in men compared with women and higher for Model B (with blood tests) compared with Model A (without blood tests) although the confidence intervals generally over-lapped" And "Sensitivity values varied considerably by cancer type with model B having higher sensitivity that model A for seven cancers (colorectal, blood, liver, lung, ovary, pancreas, prostate) and similar values for the rest".

We have added a sentence to the first paragraph of the discussion. "However, there was an improvement in sensitivity for Model B (with blood tests) compared with Model A (without blood tests) for seven cancers, there was little difference in discrimination and only a small improvement in net benefit".

3. A huge number of prediction model performance parameters are presented and it is rather difficult to see the wood for the trees. It seems to me that the key parameters are cancer site-specific sensitivity and specificity at a given cancer probability threshold. I.e. what is the performance of the model under typical scenarios in which it might be used. A cancer probability of 3% or greater is deemed clinically relevant by NICE and I think Supp Table 10 is the results table that is of major interest clinically. The overall sensitivity for detecting any cancer is not of much interest because what is needed is the probability of a specific cancer so appropriate investigations can be carried out in order to make the diagnosis.

RESPONSE: Thank you for these comments. We agree that Supp Table 10 is key and have therefore moved it to the main tables as a new Table 1 and updated the text accordingly.

Minor comments

4. Line 205 and following. The text should indicate the overall calibration (predicted numbers of cancer compared to observed) with some description of how that varied by specific cancer. A summary of the other key calibration parameters of interest (calibration intercept and calibration slope) should also be reported in the main text, not just the tables and figures. This should include some interpretation of the various parameters.

RESPONSE: We agree and have added the following text to the results, which now reads "Extended Table 6 and Extended Figures 5 and 6 show that Model A and Model B were well calibrated in the England validation cohort overall and for each cancer type in men and women. For example, the calibration intercept for Model A for lung cancer in women was close to zero with a value of -0.07 (-0.11 to -0.33) with -0.02 (-0.06 to 0.02) for men. The calibration slope was 1.00 (0.97 to 1.03) for women and 0.99 (0.96 to 1.01) for men".

5. Line 217. Decision curve analysis is not commonly used in the reporting of prediction models, and I think a more detailed explanation of what the numbers mean and how to interpret them would be helpful. Given that the management decisions are cancer site specific this part of the paper needs to be cancer site specific. A decision curve analysis related to any cancer makes little sense to me.

RESPONSE: We have added the following text to the methods *“Decision curve analysis calculates a clinical ‘net benefit’ for one or more prediction models in comparison to default strategies of treating all or no patients. It is calculated over a range of threshold risks (probabilities)”*. We have added the following interpretation to the results which now reads *“The decision curves in Figure 5 and Supplementary Figures 12-21 indicate a very small increased net benefit of using Model B compared with Model A and a greater increase compared with the Qcancer model. This means there is a similar clinical utility for Model B compared with Model A and a greater clinical utility for both models compared with a strategy of ‘treating’ everyone or non-one”*.

We have presented decision curves for any cancer and the individual cancer types. The ones for any cancer are designed to show the overall clinical utility of using the models.

We have also corrected the labels on figure 5 (as both had been labelled as women rather than one for women and one for men).

6. Line 240. Again the main text needs some summary numbers rather than simply referencing a supplementary table. It ought to be possible to read and understand the key points of a manuscript without referring to the supplement at all.

RESPONSE: Thank you for this suggestion. We have added the following text to the reclassification section of the results *“There were 8311 men (0.6% of 1,360,169) who were reclassified down and 59,103 (13.4%) who were reclassified up with Model A compared with Qcancer. The corresponding figures for women were 3736 (0.3%) and 87446 (6.9%).”*

7. Line 436. The methods for combining the various validation parameters across the multiple imputed data sets needs to be described.

RESPONSE: We have added that we combined the c statistic results across imputed datasets using Rubin’s rules and clarified that other validation statistics were run on a single imputed dataset.

8. Extended figure 5. The description of how these curves were derived need to be clearer. If they are based on the regression of the mean predicted risk against observed within risk centiles then the individual points need to be plotted (as standard calibration plots) (may also include 95% confidence intervals) in order to demonstrate the actual calibration at different risk estimates.

RESPONSE: As requested, we have replaced extended figure 5 with a scatter plot showing the individual data points for each of the centiles for models A and B instead of the lowest smoothed plot originally presented.

REFERENCES

1. Hippisley-Cox, J. & Coupland, C. Symptoms and risk factors to identify men with suspected cancer in primary care: derivation and validation of an algorithm. *BJGP* **63**, 1-10 (2013).
2. Hippisley-Cox, J. & Coupland, C. Symptoms and risk factors to identify women with suspected cancer in primary care: derivation and validation of an algorithm. *BJGP* **63**, 11-21 (2013).
3. Liao W, C.C., Burchardt J. Baldwin DR, Collaborators of the DART initiative, Gleeson FV, Hippisley-Cox J. Predicting the future risk of lung cancer: development, internal and external validation of the CanPredict (lung model) in 19.67 million people and evaluation of model performance against seven other risk prediction models. *Lancet Respiratory Medicine* [https://doi.org/10.1016/S2213-2600\(23\)00050-4](https://doi.org/10.1016/S2213-2600(23)00050-4)(2023).
4. Hippisley-Cox, J., Mei, W., Fitzgerald, R. & Coupland, C. Development and validation of a novel risk prediction algorithm to estimate 10-year risk of oesophageal cancer in primary care: prospective cohort study and evaluation of performance against two other risk prediction models. *The Lancet Regional Health - Europe* **32**(2023).
5. National Institute for Clinical Excellence. Suspected cancer: recognition and referral. NICE guideline [NG12]. Vol. 1 378 (NICE, London, 2015).
6. van Smeden, M., Reitsma, J.B., Riley, R.D., Collins, G.S. & Moons, K.G. Clinical prediction models: diagnosis versus prognosis. *J Clin Epidemiol* **132**, 142-145 (2021).
7. Vickers, A.J., van Calster, B. & Steyerberg, E.W. A simple, step-by-step guide to interpreting decision curve analysis. *Diagnostic and Prognostic Research* **3**(2019).

Manuscript: NCOMMS-24-30307-T "Development and external validation of new cancer risk algorithms to improve early diagnosis of cancer in 12.8 million people in primary care"

AUTHORS RESPONSE TO REVIEWER COMMENTS 19.09.2024

Reviewer #1 (Remarks to the Author):

I wish to thank the authors for their reply to our comments. I still believe the clarity on the purpose of the model and the risk vs probability is still unresolved and the impact of the overall probability of cancer unclear.

RESPONSE: Thank for these comments. We have now amended the manuscript to refer to probability instead of risk throughout and added additional text to clarify the purpose of the model and the impact of the overall probability of cancer. The text in the discussion reads "Currently available QCancer risk algorithms are used in clinical practice in the UK, supported by national guidelines⁷. These recommend urgent referral when the overall probability (positive predictive value) of a patient having a current as yet undiagnosed cancer based on individual symptoms is over 3% and hence prediction tools which can predict overall probability of a cancer have a utility in this context".

We have also added an additional sentence and reference to the discussion which now reads "Similarly, the overall probabilities of cancer, particularly for those with non-site-specific symptoms could help determine which patients might benefit from the non-specific cancer diagnostic pathways²⁴".

Finally, the discussion now states "As stated above the models are not intended to be used for screening for a cancer which might develop in the future over a longer period of time (such as five or ten years)".

Reviewer #2 (Remarks to the Author):

The authors have chosen not to change the language in which the term risk is used to mean a probability without any notion of time in the denominator. They cite others who have used this term in the same way. The fact that others have adopted what in my view is a poor practice is no defence. I think that greater specificity in the use of scientific terms aids understanding. The first comment of reviewer #1 is a perfect illustration of the confusion caused by the language used ("I could not find the time horizon for the prediction and this is of paramount importance."). This confusion would not have arisen if the term risk was not used to mean a simple probability. The very first sentence of the abstract "Cancer risk prediction algorithms are used in the UK to identify individuals at high risk of having a current, as yet undiagnosed cancer" illustrates the issue nicely as "Cancer risk prediction algorithms are also used in the UK to identify cancer-free individuals at high risk of developing cancer in the future". The changes required in this manuscript would be small. However, I accept that my opinion is no gold standard.

RESPONSE: Thank for these comments on which we have carefully reflected. In response,

we have now amended the manuscript to refer to probability instead of risk throughout.

I agree with the reviewer that the coefficients ought to be provided. I do not think it is acceptable to hide code behind an academic license. Especially code based on data generated using publicly funded resources.

RESPONSE:

The software which implements the algorithm is available here under an academic license for the reviewer recommended by Oxford University Innovations.

URL: <https://canpredict.org/ED/academic-use-source-code/>
username: reviewer
password: theShipLookedQuiteNormal

When the paper is published, there can be a mechanism to access this via OUI online store.
<https://process.innovation.ox.ac.uk/software/>

In other respects, the authors have addressed my comments and those of the other reviewer.

RESPONSE: Many thanks to both reviewers for their constructive comments.